# Targeted Molecular Detection of Nosocomial Carbapenemase-Producing Gram-Negative Bacteria—On Near- and Distant-Patient Surfaces

**DOI:** 10.3390/microorganisms9061190

**Published:** 2021-05-31

**Authors:** Claudia Stein, Isabel Lange, Jürgen Rödel, Mathias W. Pletz, Frank Kipp

**Affiliations:** 1Institute for Infectious Diseases and Infection Control, Jena University Hospital, Am Klinikum 1, 07747 Jena, Germany; Isabel.Lange@med.uni-jena.de (I.L.); mathias.pletz@med.uni-jena.de (M.W.P.); Frank.kipp@med.uni-jena.de (F.K.); 2Institute of Medical Microbiology, Jena University Hospital, Am Klinikum 1, 07747 Jena, Germany; Juergen.roedel@med.uni-jena.de

**Keywords:** molecular detection, fomites, carbapenemase-producing Gram-negative bacteria, hygiene, patient’s room

## Abstract

Background: Here, we describe an integrative method to detect carbapenemase-producing Gram-negative bacteria (gn-Cp) on surfaces/fomites in the patient environment. We examined environmental samples from 28 patient rooms occupied with patients who were proven to be colonised with gn-Cp by rectal screening. Methods: We took samples after 24 h, 72 h and one week. For sampling, we divided the patient environment into four parts and took samples from near- and extended patient areas. To obtain a representative bacterial swab from a larger surface, such as the patient cabinet, we used Polywipes. Bacterial DNA was isolated. Carbapenemase was detected with specific qPCR primers. Results: With this culture- and molecular-based approach, we could control the effectiveness of cleaning and disinfection in everyday clinical practice. Therefore, we could track the spread of gn-Cp within the patient room. The number of positive detections fluctuated between 30.5% (mean value positive results after 72 h) and 35.2% (after 24 h and one week). Conclusion: The method used to detect multidrug-resistant bacteria in the environment of patients by using Polywipes^TM^ is reliable and can therefore be used as an effective, new tool in hygiene and infection control.

## 1. Introduction

The fact that surface contamination and transmission by fomites play a major role in healthcare-associated infections (HAIs) is evident from many studies [1]. Multiple studies have also verified that patient rooms are poorly cleaned during terminal preparation [2], as well as terminal disinfection procedures are not studied sufficiently with regard to their efficiency and effectiveness [3] or even show an inefficiency due to insufficient levels of germicidal levels of disinfection products in use [4,5]. Nosocomial pathogens persist in high-touch, that means near-patient surfaces as well as low-touch environments and may foster an exchange of antimicrobial resistance-conferring plasmids [6]. The hospital setting provides broad possibilities for outbreaks and transmission that call for prevention strategies [7]. According to German guidelines [8], within the clinical setup, high-risk patients (patients who have recently had contact with the health system in countries with endemic occurrence; patients who have had contact with multidrug-resistant Gram-negative bacteria (MDR-gn)-carrying patients) are actively screened and isolated in single-patient rooms until MDR-gn colonisation is ruled out. In particular, the dramatic increase in the prevalence of infections caused by Gram-negative bacteria, including *Enterobacteriaceae*, as well as nonfermenters, such as *Pseudomonas aeruginosa*, producing carbapenemases (gn-Cp) is a global health problem [9]. The gn-Cp acquisition rate of 3.2% among close contacts sharing a multi-patient room (depending on the carbapenemase enzyme) shows the relevance of nosocomial transferability from patient to patient [10]. However, what is the danger from transmission through fomites? For example, a study by Weber et al. shows that gn-Cp from near-patient surfaces in rooms are cultured in only small numbers and survive [11]. In many publications, the sink is described as a source of nosocomial infections [12,13,14].

There is a substantial need for research regarding the prevention of fomite transmission, and new methods and antimicrobial materials are being developed to detect and minimise HAI. For example, much research has been done in the field of material science, such as surfaces deposited with nanomaterials [15], no-touch disinfection methods such as ultraviolet light [2], design of hospital equipment and innovative sanitation strategies based on the use of probiotic bacteria [16,17]. Novel disinfection methods, such as ultraviolet light, have been successfully used in everyday clinical practice to decrease the risk of acquiring *C. difficile* [18] and other HAI [19].

However, there are very few developments in the field of fomite transmission detection. A practical and reliable detection system for microorganisms is needed for cleaning quality control, the establishment and validation of new disinfection strategies and outbreak investigations.

Detection methods of bacteria and their resistances are improved through rapidly developing technologies in molecular diagnostics, such as whole-genome sequencing (WGS), and consequently advance the therapy of patients with HAI. The challenge now is to apply these methods to the detection of fomites. Microbial contamination on near-patient surfaces can easily be transferred from a colonised to a susceptible patient and are recognised as a source of HAI [20]. The bacteriological examination of near-patient surfaces offers a starting point to interrupt the nosocomial transmission chain at an early stage.

Here, we describe a flexible method to detect gn-Cp and examine their distribution in and around the patient room.

## 2. Materials and Methods

### 2.1. Sampling Strategy in the Patient Area

From December 2018 to June 2020, we examined environmental samples from patient rooms occupied with patients admitted to our 1400-bed hospital with a positive result in nosocomial screening swabs for gn-Cp and included these patients in our study. The sampling for this study is part of our multidrug-resistant pathogen-monitoring system. At our hospital, we sequenced all gn-Cp isolates, regardless of whether colonisation or infection occurred. The clonality is determined to record any nosocomial transmission. If gn-Cp colonisation is detected, the patient is cared for in a single patient room, as is recommended for patients with gn-Cp colonisation or infection (by the German guidelines) [8]. In the patients’ room, we examined the environment by probing different surfaces at different times. Depending on the duration of the patient’s hospitalisation, we took samples after 24 h, 72 h and one week. For sampling, we divided the patients’ environment into four parts. Parts and surfaces in the **immediate patient surroundings with direct hand contact** were grouped (bed rails, keypads left and right, multifunction centre with remote control, telephone and bedside table: only surface and handles as **area ‘hand’**). The **extended patient environment** (no hand contact) was pooled in an observation group (bed frame bottom, patient coat closet and deep inside as **area**
**‘extended’**). The last group summarised surfaces in the **wet room of the patient** (e.g., WC flushing (push button) and faucet; **area**
**‘****wet****’**). We took eight samples in and around each patient room. Three were taken from area ‘hand’, two from area ‘extended’ and one from area ‘wet’. **Outside of the patient room**, two additional samples were taken (**area**
**‘****outside****’**). One sample was taken from the outside of the door and the handrail (approximately 50 cm). Another sample was taken from the nurses’ workroom for dismissing potential infectious fluids (control panel of the bedpan sink).

Surfaces of area ‘hand’, area ‘wet’ and area ‘outside’ are cleaned once a day with the routine surface disinfectant Mikrobac^®^ forte 0.5%. The extended patient environment (area ‘extended’) is only disinfected if the patient is moved to another room or discharged, but is not part of the daily routine cleaning.

### 2.2. Sampling on Surfaces

To sample a large surface, such as the patient cabinet, we used Polywipes^TM^ (medical wire, Corsham, Wiltshire, UK). These buffered wipes take up microbes from a much larger area than simple swabs. Sterile collection is important for sampling. Two infection control practitioners carried out the sampling. With a sterile glove, the wipe was removed from the sample package, and the surface to be tested was wiped off. After removal, the wipe was placed back in the sterile transport box. The wipes are very well suited to test larger surfaces as they absorb more material and have a larger surface than simple swabs. The buffered dishes can be stored overnight at 4 °C without affecting the result.

### 2.3. Enrichment and Incubation

To enrich the bacteria within the wipe, 50 mL of CASO boullion (tryptic soil broth with neutralizers, Merck KGaA, Darmstadt, Germany) was added to each cloth. CASO broth is particularly well suited as a collective nutrient medium for incubating microorganisms collected from cleaned or disinfected surfaces. The contained detergents (polysorbate 80, Na-thiosulfate, cysteine, histidine and lecithin) inactivate any present disinfectants, which can lead to the inhibition of bacterial growth. The boullion was aerobically incubated at 37 °C and 160 rpm for 16 h. One hundred microlitres of the turbid boullion was plated on MH agar supplied with 2 mg/L meropenem (TCI Deutschland GmbH, Eschborn, Germany), to inhibit carbapenem-sensitive attendant flora. As a growth control, a carbapenem-positive isolate, as well as a carbapenem-negative isolate, was used. The agar plates were incubated overnight at 37 °C.

### 2.4. DNA Isolation and PCR

Viable cells capable of reproduction are used for the DNA isolation and the subsequent PCR. Three to five colonies from the MH agar plate were used for DNA isolation. Therefore, a NucleoBond^®^ AXG20 DNA extraction kit (Machery Nagel GmbH & Co. KG, Düren, Germany) was used.

The isolated DNA was stored until molecular biological evaluation in DNA/RNA-free water (Life Technologies GmbH, Darmstadt, Germany) at 4 °C. For each sample of bacterial growth on MH agar, PCR was performed specifically for the carbapenemase detected in the rectal sample of the patient. As a control, carbapenemase-positive and carbapenemase-negative samples were subjected to PCR. Additionally, a no-template control was used to identify PCR contamination. Real-time PCR was performed in a Rotor-GeneQ cycler (Qiagen, Hilden, Germany) by directly applying 2 µL of the isolated DNA. The PCR mixture was composed of 1.8 mM MgCl_2_, 1× PCR buffer (Life Technologies GmbH, Darmstadt, Germany), 0.2 mM dNTPs (Roth, Karlsruhe, Germany), 5 µM each primer (Sigma Aldrich, Munich, Germany; Table 1), 0.15 x SYBR Green (Life Technologies GmbH, Darmstadt, Germany), 0.08 U/µL Platinum Taq DNA Polymerase (Life Technologies GmbH, Darmstadt, Germany) and 0.1 mg/mL BSA (Life Technologies GmbH, Darmstadt, Germany). The PCR was run as follows: pre-denaturation at 99 °C for 10 s and 95 °C for 50 s, followed by 45 cycles composed of 95 °C for 20 s, annealing for 20 s (see Table 1) and 72 °C for 20 s. The melting temperatures of PCR products were determined by increasing (0.5 °C/4 s) the temperature stepwise (from 75 °C to 99 °C).

Based on the melting temperature of the PCR product of the positive control, samples of the wipes were evaluated. We only evaluated samples if all controls were error free. With the PCR primers used, specific PCR products could be generated and used for sample evaluation. An overview of the method is displayed in Figure 1.

## 3. Results

### 3.1. Sampling in the Patient Area

A total of 28 patients, colonised with gn-Cp, were included in this study and thus 28 patient rooms were examined. Overall, 529 samples were collected within the hospital stay. Growth on the meropenem plates was seen in 63.3% (335/529). Of these, 34.6% (183/529) showed a specific amplification product for the investigated carbapenemase according to the PCR analysis.

From the 28 patients colonised with gn-Cp, we detected three colonised with *Pseudomonas aeruginosa*, ten with *Citrobacter freundii*, eight with *Escherichia coli*, two with *Enterobacter cloacae*, three with *Klebsiella pneumoniae*, one with *Klebsiella oxytoca* and one with *Enterobacter hormaechei*. The carbapenemases found were 16 VIM, five OXA-48, five NDM, one IMI and one KPC.

### 3.2. Spread of Multidrug-Resistant Gram-Negative Pathogens with Carbapenemase after 24 h

After 24 h, we found the lowest proportion of carbapenemase-PCR positive samples in the tested patient rooms at the multifunctional centre. The proportion of positive results averaged between 36 and 37% for the four individual areas (Figure 2). The area ‘outside’ the patient room was slightly less exposed (33%) than the area ‘hand’ near the patient (mean area ‘hand’, 36.5%). Overall, we found the least positive PCR evidence in the unclean workroom (28.6%).

### 3.3. Spread of Multi-Resistant Gram-Negative Pathogens with Carbapenemase after 72 h

The number of positive results for all sampling points was lower on average after 72 h (mean value, 30.5%) than after of 24 h (mean value, 35.2%). The sampling location bed frame bottom in the area ‘extended’ (area ‘extended’ mean value, 40%) was the most heavily burdened area (Figure 3). The load with carbapenemase-producing organisms was significantly higher inside the patient’s room (mean value, 33%) than in the tested areas outside the room (mean value, 25%).

### 3.4. Spread of Multi-Resistant Gram-Negative Pathogens with Carbapenemase after One Week

Overall, the number of positive detections after the first week (mean value positive results, 35.2%) raised again to the level at 24 h (mean value positive results, 35.2%) after the slight decrease after 72 h (mean value positive results, 30.5%).

We found less positive evidence in the area outside the room (area ‘outside’ mean value, 22.7%) than in the room interior (mean values, 45.5% and 39.4%). In the examinations after one week, we found the highest exposure at all time points within the patient’s room. The positive rate of 45.5% at the measuring points bed frame bottom, patient coat closet, bed rails and bedside table was a very high level of exposure of living microorganisms, which are capable of reproducing (Figure 4).

### 3.5. Spread of Multi-Resistant Gram-Negative Pathogens (Independent of Time)

The sampling points outside the patient room showed the fewest positive findings. Interestingly, there was no difference in the exposure to gn-Cp between the areas with hand contact (area ‘hand’ mean value positive results, 27.4%) and areas without hand contact (area ‘extended’ mean value positive results, 28.9%), although the disinfection cycle also differs in these areas. If we look at the samples at all times, we had the most positive PCR results on the patient’s bed frame. It is also noticeable that the area ‘wet’ (mean value positive results, 21.4%) was under the load of the remaining tested areas in the room (Figure 5).

## 4. Discussion

Here, we present a structured and precise combined culture- and molecular-based approach to screen surfaces for gn-Cp. Often, new and expensive equipment must be purchased to establish new detection techniques. The advantage of this method is the combination of already established technologies and is that it is thus feasible for every laboratory. The only devices needed are an incubator and a PCR machine. Noteworthy, this method detects resistance genes from viable bacteria, as the process is based on a preceding culture-based enrichment of the microorganisms in the CASO-bouillon followed by selective agar plates supplied with meropenem. The focus of detection, depending on the requirements, may be ESBLs, carbapenemases, VRE, or other mechanisms of multidrug resistance.

Adding PCR as a detection tool for resistance genes, the method becomes much more specific since a large number of phenomena can lead to growth on carbapenem selective agar plates in the absence of carbapenemases, such as *Pseudomonas aeruginosa* with an intrinsic efflux mechanism or numerous *Enterobacteriaceae* with ESBL and porin loss [25,26]. Even if discrimination between different mechanisms of carbapenem resistance may not be relevant from the perspective of the treating physician, it has indeed implications regarding infection control. Gn-Cp have a much higher outbreak potential compared to the carbapenem-resistant Gram-negative bacteria that use alternative mechanisms of resistance, probably because efflux and porin loss is associated with a higher fitness cost. Therefore, detection of gn-Cp is of high relevance from an infection control perspective [27].

The major difference from other investigations regarding fomites is that we did not sample with a swab or contact plates (replicate organism detection and counting, RODAC) but with a sponge soaked with buffer. The swab sample is useful for small areas, such as a sink, but sampling with a swab, especially when it is dry, is not representative for larger areas [28].

Because of its convenient implementation in practice and reliable performance, the methodology is an accessible tool for clinical routine use. Therefore, screening to control the effectiveness of cleaning and disinfection methods is possible. For the convenient and increasingly widely used disinfectant-impregnated wipes, validation studies on the disinfection efficacy in clinical practice are needed [29]. This strategy makes it very easy to test new cleaning and disinfection strategies, such as novel photodynamic coatings, in everyday clinical practice [20]. Improved cleaning strategies of room surfaces decrease the risk of fomite-transmitted HAI [30,31]. Furthermore, nosocomial transmission chains can be identified. It must be noted that the detection of fomites and the implementation of new methods for interrupting infection chains can be successful only if the awareness is truly high among hospital staff [32].

With the method presented here, we focused on patient rooms occupied by patients confirmed to be colonised with gn-Cp. Comparable to our results were the data from Shams et al., who also collected environmental samples with a sponge and obtained a contamination rate of 34% total bacteria of multidrug-resistant organisms within the patient room [33]. Compared to other studies [11,34], the detection yield of gn-Cp with an average of 34.5% in all surface samples (183/530) is very high. One reason can be an increase of sensitivity due to the use of a sponge covering larger areas compared to traditional swabbing or contact plates. In the studies from Weber et al. and O’Fallon et al., for example, the test field of the environmental sample was approximately 5 × 5 cm and thereby limited to the size of the contact plates [34]. Therefore, comparing our results to other studies using traditional methods is limited. Unfortunately, Rock et al. made no statement about the recovery of gn-Cp from sponges compared to that from the swabs [35].

It was notable that we measured the highest levels of contamination after one week. Obviously, the daily routine cleaning of the room did not achieve a sufficient reduction in contamination. This finding also coincides with the results of the study by Shams et al., where 45% of routinely cleaned rooms and 30% of terminally cleaned rooms had positive multidrug-resistant pathogens. However, we were also able to record a reduction after 72 h.

The difference in gn-CP detection rates between immediate patient surroundings with direct hand contact and the extended patient environment was low. This result is in contrast to other studies, in which the areas closest to the patient were usually the most contaminated [33,36]. Our findings may result from the fact that the area ‘extended’ is not included in the daily cleaning and disinfection procedures. Furthermore, the role of the transmission through fomites with the extended patient area has not been clarified.

Additionally, in the area where many faecal pathogens are suspected, as in the wet area, we found fewer organisms. One could speculate that manual cleaning and disinfection was carried out more thoroughly in this area because a greater need for cleaning was expected. The fact that we were able to detect gn-CP outside the room on the handrails is problematic from a hospital hygiene point of view. It can be a sign that hand hygiene is still in need of improvement. Regarding these findings, our results indicate that precise detection tools are needed as a basis to prevent nosocomial transmission of gn-CP.

## 5. Conclusions

The method used to detect multidrug-resistant bacteria in the environment of patients by using Polywipes^TM^ is reliable and can therefore be used as an effective, new tool in hygiene and infection control. In our study, we examined several areas in and around the patient’s room for contamination with multi-resistant Gram-negative pathogens. The number of positive detections fluctuated between 30.5% and 35.2%. Our results confirm the role of patient-side contamination in nosocomial transmission and subsequent infections. For infection control it is crucial to detect potential sources of environmental contamination. Precise detection tools like these are needed as a basis to prevent nosocomial transmission and outbreaks, especially with Gram-negative bacteria.

## Figures and Tables

**Figure 1 microorganisms-09-01190-f001:**
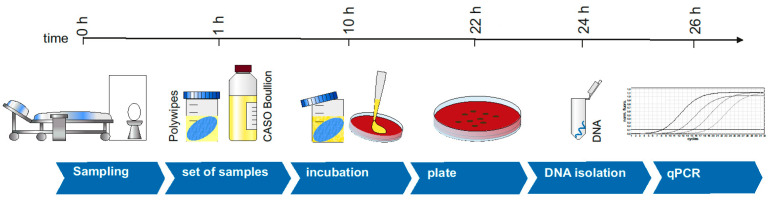
Study design. Schematic overview of the individual steps in the methodology.

**Figure 2 microorganisms-09-01190-f002:**
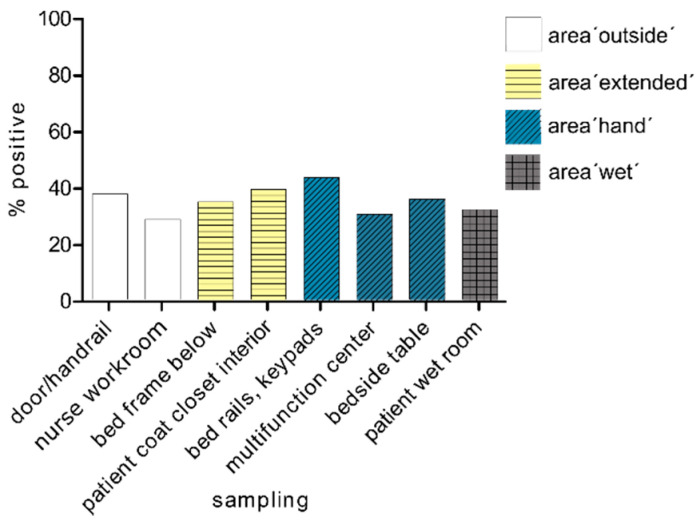
Spread after 24 h. Spread of carbapenemase-positive organisms within the four tested areas after 24 h of hospitalisation of the patient in the room (*n* = 21–30).

**Figure 3 microorganisms-09-01190-f003:**
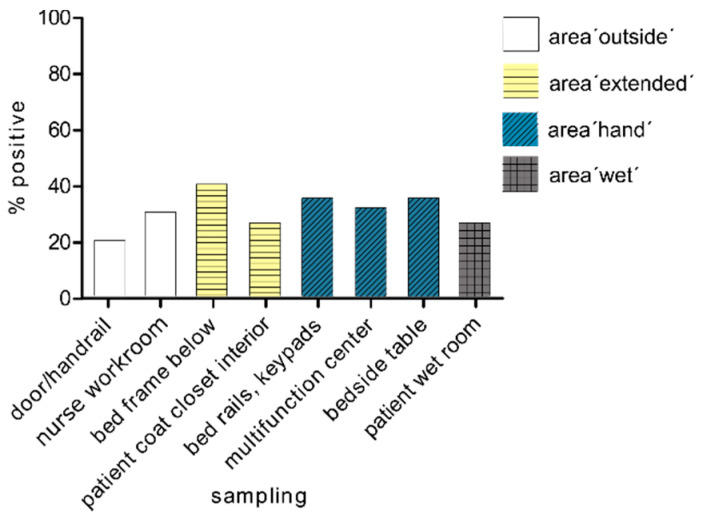
Spread after 72 h. Spread of carbapenemase-positive organisms within the four tested areas after 72 h of hospitalisation of the patient in the room (*n* = 19–20).

**Figure 4 microorganisms-09-01190-f004:**
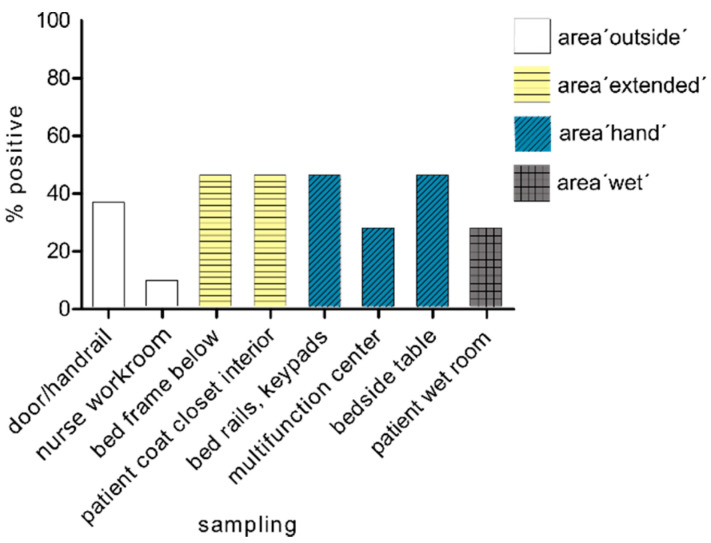
Spread after one week. Spread of carbapenemase-positive organisms within the four tested areas after one week hospitalisation of the patient in the room (*n* = 11).

**Figure 5 microorganisms-09-01190-f005:**
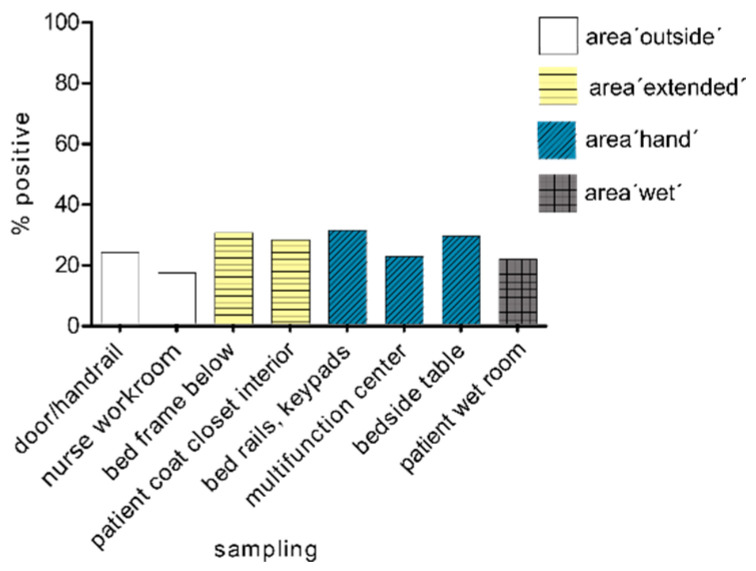
Spread of carbapenemase-positive organisms within the four tested areas after 24 h, 72 h and one week of hospitalisation of the patient in the room (*n* = 51–61).

**Table 1 microorganisms-09-01190-t001:** PCR primer sequences for the analysed carbapenemases.

Carbapenemase	Forward Primer	Reverse Primer	Annealing Temperature	Reference
VIM	TGGCAACGTACGCATCACC	CGCAGCACCGGGATAGAA	61 °C	[21]
OXA-48	GCGTGTATTAGCCTTATCGGCTG	GCGGGTAAAAATGCTTGGTTCGC	60 °C	[22]
IMI	ATAGCCATCCTTGTTTAGCTC	TCTGCGATTACTTTATCCTC	62 °C	[23]
NDM	GGTTTGGCGATCTGGTTTTC	CGGAATGGCTCATCACGATC	68 °C	[24]
KPC	CGTCTAGTTCTGCTGTCTTG	CTTGTCATCCTTGTTAGGCG	68 °C	[24]

## Data Availability

All relevant data are within this article.

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
