# Peer review of "Targeted Molecular Detection of Nosocomial Carbapenemase-Producing Gram-Negative Bacteria—On Near- and Distant-Patient Surfaces"

_microorganisms, 2021, doi:10.3390/microorganisms9061190_

Round 1

Reviewer 1 Report

The manuscript of Stein and co-authors (Targeted molecular detection of nosocomial carbapenemase-2 producing gram-negative bacteria - on near- and distant-patient 3 surfaces.) documents the analysis of occurrence of gram negative bacteria producing carbapenemase-2 using PolywipesTM. For the following analysis PCR methods were used.

Here they show the quick and reliable detecting of those bacteria in a hospital area. The experiments in general are well documented and discussed.

However, table 1 is a little bit confusing. The abbreviations are not explained, so it is difficult to follow there meaning. A look in the reference list does not help, since all the mentioned references are not listed there.

Therefore manuscript should be carefully checked for the references.

Author Response

Author's Reply to the Review Report (Reviewer 1)

Point 1: However, table 1 is a little bit confusing. The abbreviations are not explained, so it is difficult to follow there meaning.

Answer author:

Unfortunately, I cannot understand the objection that the abbreviation was not explained. Beta lactamases are usually not written out in full but abbreviated. I can, however, ask the editor how it is handled in the journal and change it if necessary because it is not mentioned explicitly in the instruction for authors.

The table seems a bit confusing due to the table style given by the journal. I have improved the formatting now. Because this is specified by the journal, I will discuss the format of the table with the editor.

Point 2: A look in the reference list does not help, since all the mentioned references are not listed there. Therefore manuscript should be carefully checked for the references.

Answer author:

I made a mistake with the reference list in the table and formatted the references not correctly. I edited the references in the manuscript (Please see the attachment).

Reviewer 2 Report

This is an interesting paper defining  an integrative method to detect carbapenemase-producing Gram-negative bacteria (gn-Cp) on surfaces/fomites in the patient environment.

However I have some comments for authors.

To change gram with Gram

line 34 add the following reference Cortegiani et al. What Healthcare Workers Should Know about Environmental Bacterial Contamination in the Intensive Care Unit. BioMed Research International 2017(4):1-7

 The conclusions are intuitive and the findings don't support an innovative discovery.  Please, insert some comment about the results obtained.

Author Response

Author's Reply to the Review Report (Reviewer 2)

Point 1: To change gram with Gram

Answer author:

I changed gram to Gram.

Point 2: line 34 add the following reference Cortegiani et al. What Healthcare Workers Should Know about Environmental Bacterial Contamination in the Intensive Care Unit. BioMed Research International 2017(4):1-7

Answer author:

I add the reference. Thanks for the note.

Point 3: The conclusions are intuitive and the findings don't support an innovative discovery. Please, insert some comment about the results obtained.

Answer author:

I have rephrased the conclusions (Please see the attachment).
